# *Mycoplasma genitalium* Infections and Associated Antimicrobial Resistance in Canada, 1980–2023

**DOI:** 10.3390/tropicalmed10050139

**Published:** 2025-05-19

**Authors:** Angela Copete, Mariana Herrera, Camilo Suarez-Ariza, Zipporah Gitau, Maria Arango-Uribe, Rotem Keynan, Camila Oda, Ameeta E. Singh, Stuart Skinner, Cara Spence, Will Riaño, Lauren J. MacKenzie, Ken Kasper, Laurie Ireland, Irene Martin, Jared Bullard, Lucelly Lopez, Diana Marin, Margaret Haworth-Brockman, Yoav Keynan, Zulma Vanessa Rueda

**Affiliations:** 1Department of Medical Microbiology and Infectious Diseases, University of Manitoba, Winnipeg, MB R3E 0J9, Canada; copetera@myumanitoba.ca (A.C.); mariana.herreradiaz@umanitoba.ca (M.H.); suarezac@myumanitoba.ca (C.S.-A.); gitauz@myumanitoba.ca (Z.G.); arangoum@myumanitoba.ca (M.A.-U.); rotykeynan@gmail.com (R.K.); odac@myumanitoba.ca (C.O.); lauren.mackenzie@umanitoba.ca (L.J.M.); yoav.keynan@umanitoba.ca (Y.K.); 2Department of Medicine, University of Alberta, Edmonton, AB T6G 2R3, Canada; ameeta@ualberta.ca; 3Department of Medicine, University of Saskatchewan, Saskatoon, SK S7N 5A2, Canada; stuart.skinner@usask.ca (S.S.); cara.spence@usask.ca (C.S.); 4Wellness Wheel Clinic, Regina, SK S4T 2M7, Canada; 5Grupo Bacterias & Cáncer, Facultad de Medicina, Universidad de Antioquia, Medellin 050010, Colombia; 6Department of Internal Medicine, University of Manitoba, Winnipeg, MB R3T 2N2, Canada; 7Manitoba HIV Program, Winnipeg, MB R3G 0X2, Canada; 8Health Sciences Centre, Winnipeg, MB R3A 1R9, Canada; 9Nine Circles Community Health Centre, Winnipeg, MB R3G 0X2, Canada; 10Department of Family Medicine, University of Manitoba, Winnipeg, MB R3T 2N2, Canada; 11Streptococcus and STI Unit, National Microbiology Laboratory, Public Health Agency of Canada, Winnipeg, MB R3E 3R2, Canada; 12Cadham Provincial Laboratory, Shared Health, Winnipeg, MB R3E 3J7, Canada; 13Department of Pediatrics and Child Health, University of Manitoba, Winnipeg, MB R3T 2N2, Canada; 14School of Medicine, Universidad Pontificia Bolivariana (UPB), Medellin 101001, Colombia; lucelly.lopez@upb.edu.co (L.L.); dianamarcela.marin@upb.edu.co (D.M.); 15Department of Community Health Sciences, University of Manitoba, Winnipeg, MB R3T 2N2, Canada; margaret.haworth-brockman@umanitoba.ca; 16National Collaborating Centre for Infectious Diseases (NCCID), University of Manitoba, Winnipeg, MB R3T 2N2, Canada

**Keywords:** Canada, *Mycoplasma genitalium*, macrolide resistance mutations, fluoroquinolone resistance mutations, scoping review

## Abstract

Background: We aimed to describe trends in *M. genitalium* prevalence and associated resistance in Canada between 1980 and 2022. Methods: Ecological study and a scoping review. We collected publicly available data published by the governments of all Canadian provinces and territories. We also systematically searched PubMed, Medline, Embase, and grey literature using the keywords ‘*M. genitalium*’, ‘Canada’, and all provinces and territories. We reported *M. genitalium* prevalence, age, sex, gender, symptoms, coinfections, sample types used for diagnosis, and macrolide and fluoroquinolone resistance rates. Results: National or provincial surveillance systems for *M. genitalium* are absent. Eight studies reported the epidemiology of *M. genitalium.* The prevalence ranged between 3% in Quebec and 30.3% in Ontario. Half of the patients reported symptoms. The most collected sample for *M. genitalium* diagnosis was urine, followed by cervical and urethral swabs. Co-infection with *Chlamydia trachomatis* was reported in 3.3% to 16.4% of cases and with *Neisseria gonorrhoeae* in 0.0% to 24.0%. Macrolide resistance ranged between 25% and 82.1%, and fluoroquinolone resistance between 0.0% and 29.1%. Conclusions: *M. genitalium* prevalence and resistance rates varied by sex, gender, province, and specimen type. In the absence of routine surveillance, incomplete data hinders understanding the bacterium’s natural history, its impact on some key groups, and the tracking of antibiotic resistance.

## 1. Introduction

*Mycoplasma genitalium* is a Mollicute class bacterium considered a causative agent of male urethritis, female urethritis, and cervicitis [1]. Long-term consequences, such as infertility, ectopic pregnancy, and pelvic inflammatory disease, have been reported for reproductive women’s health [2]. The impact of chronic infections in males is less clear [2,3,4].

The global epidemiology of *M. genitalium* infection remains unknown due to the lack of regular testing and bacterial surveillance [5,6,7,8,9]. It is suggested that prevalence tends to be higher than *Neisseria gonorrhoeae* and similar to *Chlamydia trachomatis* in several key populations, such as sex workers, men who have sex with men, people who attend care at sexual clinics, and in low- and middle-income countries [1,10,11,12]. In addition, rate variations have been reported according to the samples used for diagnosis in females and males [7,9]. Approximately half of the patients do not exhibit symptoms. Therefore, it is difficult to estimate the true prevalence and the carrier state contributing to transmission [13,14,15].

Unfortunately, rates of antibiotic resistance to traditional empiric therapies have increased in recent years. Macrolides are used for syndromic therapy of urethritis; hence, individuals are frequently exposed to this class prior to treatment of *M. genitalium*. Macrolides, initially recommended as the first line of treatment, are now threatened due to over 50% macrolide resistance in *M. genitalium* isolates in Canada [16] and 80% in other countries [17]. In addition, the global resistance to fluoroquinolones in *M. genitalium* is 13.3% [18], with differing prevalence in some geographic areas (8.4% in Europe vs. 35.6% in the Western Pacific) and in some key populations such as people living with HIV, people with recent sexually transmitted infections (STIs) diagnoses, or people who attend STI clinics [18,19,20,21,22].

*M. genitalium* infections are not notifiable infectious diseases in Canada [23]. Canadian guidelines do not recommend routine testing for *M. genitalium*. International guidelines recommend *M. genitalium* screening only for persons with persistent symptoms or recurrent episodes of urethritis, cervicitis, and pelvic inflammatory disease (PID) when chlamydia and gonorrhea have been ruled out as the etiological cause of these syndromes [6,7,9,24]. Access to testing is limited to cases of recurrent cervicitis or urethritis with negative results for *Chlamydia trachomatis* and *Neisseria gonorrhoeae*. Currently, testing for *M. genitalium* diagnosis is mostly available in provincial laboratories, and the molecular antimicrobial resistance testing is available by referring clinical specimens to the National Microbiology Laboratory in Winnipeg, Canada [23]. Knowledge of the current prevalence of both asymptomatic and symptomatic infections and the rates of antimicrobial resistance is scarce.

Population-based *M. genitalium* data are needed to understand the prevalence of *M. genitalium*, risk factors and drivers of development of resistance, and its implications for treatment [17,25]. Reviewing national and regional data is crucial to understanding local variations and geographical distribution and the impact of *M. genitalium* over time, to identifying which groups are disproportionately affected, and to prioritizing resource allocation.

In the present study, we summarize the publicly available information on *M. genitalium* infections and resistance patterns in Canada and its provinces. We aimed to describe the rates of infection, macrolide and quinolone resistance in different population groups, and the sample types used for *M. genitalium* diagnosis.

## 2. Materials and Methods

### 2.1. Study Design

Ecological study combined with a scoping review.

### 2.2. Study Questions

What is the prevalence, incidence, or frequency of *M. genitalium* in Canada (by provinces and territories, sex and gender, and specimen type)? What are the most common diagnostic tests used to detect *M. genitalium*? What are the macrolide and fluoroquinolone resistance frequencies?

### 2.3. Data Collection

We (seven authors and the librarian) systematically searched publicly available data reported between 1980 and 2023 published by the federal Canadian surveillance system and the surveillance systems of the governments of the 10 Canadian provinces and three territories.

In addition, for the scoping review, we systematically searched three major electronic databases (PubMed, MEDLINE, and Embase) and grey literature sources of information published between 1980 and 2023. The search strategy included the following keywords: ‘*M. genitalium*’, ‘Canada’, and the name of each Canadian province (‘British Columbia’, ‘Alberta’, ‘Saskatchewan’, ‘Manitoba’, ‘Ontario’, ‘Quebec’, ‘Nova Scotia’, ‘Newfoundland and Labrador’, ‘New Brunswick’, and ‘Prince Edward Island’) and territories (‘Yukon’, ‘Nunavut’, ‘Northwest Territories’).

We used the following inclusion criteria: reports or articles documenting the incidence or prevalence of *M. genitalium* in the Canadian population or its provinces and territories. We also included peer-reviewed manuscripts that reported *M. genitalium* antibiotic resistance tested among patients or participants.

We excluded studies that focused exclusively on in vitro or diagnostic test development or other laboratory developments, validation, or standardization. Case reports and literature reviews were also excluded, although we reviewed the references of those papers to identify additional publications not retrieved by the search strategies.

Two reviewers independently screened and blinded the titles and abstracts as well as the full-text papers. Discrepancies were resolved by consensus with a third reviewer. The quality of each included study was evaluated by two reviewers in an independent and blinded manner, using the Quality Assessment Tool for Observational Cohort and Cross-Sectional Studies of the National Institutes of Health (NIH), United States [26].

### 2.4. Variables

We collected *M. genitalium* prevalence, incidence, or frequency in general and reported by province/territory, age, sex, gender, and specimen type used for diagnosis. In addition, we extracted information on the year, population, and location where the study was conducted; sexual orientation; ethnicity; symptoms; prevalence or frequency of coinfections with *Chlamydia trachomatis* and *Neisseria gonorrhoeae*; the prevalence or frequency of resistance to macrolides and fluoroquinolones; and the most critical genomic mutations conferring the resistance patterns.

### 2.5. Statistical Analysis

Descriptive statistics were used to summarize sociodemographic characteristics. We reported the prevalence or frequency of *M. genitalium* infections and molecular antibiotic resistance to macrolides and fluoroquinolone antibiotics. In addition, we reported the proportion of *M. genitalium* diagnoses by age, sex, gender, sexual orientation, specimen types used for clinical diagnosis, and ethnicity.

## 3. Results

We did not find any Canadian or provincial/territorial surveillance systems. Between 1980 and 2025, a provincial report of *M. genitalium* antimicrobial resistance was found [27].

The systematic search identified 65 titles, of which 13 were duplicates. The remaining 52 abstracts were screened, two additional duplicates were removed, and 42 were excluded as described in the flowchart (Figure 1). In the end, eight studies were included. The quality assessment is described in the Appendix A.

The number of individuals included in these studies ranged from 198 to 2294 [28,29,30,31,32,33,34,35], and the overall prevalence ranged between 3% and 30.3% (Table 1).

Overall, females/women in the research ranged from 16 to 46 years old (median or mean 25–27 years old), and males/men from 16 to 74 years old (median or mean 30–32 years old).

Only three studies disaggregated data by sex, with prevalence between 3.2% [33], 7.2% [29], and 11% [32] in females and 4.5% [33], 5.3% [29], and 6.2% [32] in males. The prevalence/frequency of *M. genitalium* by gender also varied, ranging from 3%, 5%, 7.4%, 11%, 16%, 20%, 19.7%, and 30.3% in women [30,31,35]; and 15.3% in men [28], and varied between 5.7% [34] and 6.6% [29] in gay, bisexual, and other men who have sex with men (gbMSM).

The prevalence of *M. genitalium* also varied by the presence or absence of symptoms. Four studies reported that the prevalence in asymptomatic people ranged between 52% [33], 87.3% [31], 86.6% [35], and 91.7% [29], and in symptomatic people between 5% [28], 8.2% [29], 10.6% [35], 12.6% [31], and 48% [33]. The frequency of symptoms varied by sex, from 40% [33] to 44.3% [29] in females and 40.5% [29] to 50% [33] in males. The symptoms reported by women were vaginal discharge, odour or itching, bleeding between periods, painful sex or urination, and for men were pain, dysuria, tingling, or urethral discharge [28,29].

Regarding ethnicity, only one study from Alberta reported the percentage of *M. genitalium* infections in Indigenous people was 15.2% (26/171), 5.1% (79/1536) in Caucasians, and 7.2% (30/417) in people from other ethnicities [29].

Another key finding was the co-detection of other sexually transmitted bacteria among people with *M. genitalium* infections. Co-infections with *Chlamydia trachomatis* varied between 3.27% (22/672), 4% (2/50), 13.4% (53/396), 16.4% (31/189), and 6.7% (4/59) [29,30,32,33,35]; and with *Neisseria gonorrhoeae* between 0.0% (0/23), 6.8% (13/189), 10% (5/50), 6.7% (4/59), and 24% (12/50) [27,30,31,32,33]. The prevalence of HIV coinfection was 2.0% (2/96) [29].

The detection rate of *M. genitalium* varied by sample type. Table 2 shows the distribution of *M. genitalium* in different specimen types. The most common specimen types were urine and cervical or vaginal swabs, followed by urethral, rectal, and pharyngeal swabs (Table 2).

Table 3 reports the macrolide and fluoroquinolone resistance patterns reported in the studies. Macrolide resistance ranges between 25% and 82.1%, and fluoroquinolone resistance ranges between 0.0% and 29.1%. Mutations at position 2058 (*Escherichia coli* numbering) of 23S rRNA ranged from 2.72% to 57.6% [28,29,30,31,32,34,35], and mutations at position 2059 ranged from 11.6% to 84.7% [28,29,30,31,32,34,35]. Mutations in the *parC* gene ranged between 1.9 and 29.1% [29,30,32,33]. Mutations in *gyrA* were not tested or not detected [27,28,32]. Detailed information about the frequency of mutations is described in the Appendix A.

## 4. Discussion

In our study, no surveillance information or data describing national population prevalence of *M. genitalium* were identified in our systematic search. The scoping review found that *M. genitalium* prevalence varied by sex, gender, province, presence and absence of symptoms, and specimen used for diagnosis. We also found that the molecular macrolide resistance varies across the provinces from 25% to 82.1%, and fluoroquinolone resistance was lower (0.0–29.1%).

Our study identified different geographic and population prevalence of *M. genitalium* infections, ranging from 3% in the province of Quebec to 30.3% in Ontario. The variation in the prevalence, or frequency, of *M. genitalium* in Canada is comparable to the worldwide distribution. Bauman et al. found in a systematic review and meta-analysis an *M. genitalium* prevalence of 0.80% (95% confidence interval (CI), 0.42–1.57) to 9.10% (95% CI, 6.10–13.50) in the general population, and up to 26.30% (95% CI, 23.30–29.40) in commercial sex workers [1]. The prevalence is also higher in some settings and key populations, e.g., 4.9% in female sex workers in Ecuador [36], 11.5% (21/182) in female sex workers in Burkina Faso [37], and 10.2% in women living with HIV in sub-Saharan Africa [38].

We found that females diagnosed with *M. genitalium* were younger (median or mean 25–27 years old) than males (median or mean 30–32 years old). This is similar to the findings in a systematic review of population-and clinic-based diagnoses of *M. genitalium*, which reported less than 2% prevalence among women and men <25 years old [1]. However, the data heterogeneity in age ranges and enrollment procedures limits our ability to draw firm conclusions. Another U.S. study showed that overall, the prevalence was higher in younger ages in both females and males [females < 30 years (19.2%; 95% CI [15.6–23.4]) versus females > 30 years (7.2%; 95%CI [3.8–13.1]); and males < 30 years (22%; 95% CI [17.5–27.4]) versus males > 30 years (9.2%; 95%CI [5.7–14.6])] [10].

The prevalence also varies by sex and gender. Several articles reported data by sex and gender [28,30,31,33], although a few used these terms interchangeably in the same report. *M. genitalium* prevalence among females ranged between 3.2% and 19.7% (women: 3–30.3%) and among males, between 4.5% and 6.2% (men: 15%). Recently, a systematic review of *M. genitalium* prevalence across different population groups did not find significant differences between women and men [1], regardless of whether the detection was carried out in the community or clinical settings. Similarly, Getman et al. did not find differences in *M. genitalium* infection distribution by sex (16.3% in females vs. 17.2% in males) in a multicenter study [10].

Emerging evidence is showing differences in *M. genitalium* bacterial load between females and males. Munson et al. found differences in median and mean titers of log10 *M. genitalium* data from males (4 and 3.67, respectively) versus females (3 and 2.98, respectively; *p* < 0.0001) [39]. In addition, Munson et al. reported that male rectal swabs exhibited a higher rRNA target burden than male first-void urine (*p* = 0.0002); the same differences were consistent for vaginal and endocervical swabs vs. first-void urine (*p* = 0.008) [39]. Further studies are needed to understand biological differences in females and males and how those differences can impact diagnostic performance in females with low bacterial density.

Canadian data disaggregated by sexual orientation are scarce. Two studies reported data in gbMSM, and the prevalence was between 5.7% (41/716) and 6.6% (79/1212) [29,34], varying by specimen types used for diagnosis. In global studies, *M. genitalium* prevalence is higher in gbMSM, and even higher in gbMSM living with HIV (41.5%) [40] when compared to the overall population [1,15]. The reported Canadian prevalence in gbMSM is lower than in other high-income countries: 9.5% (95/1001) in asymptomatic MSM in Australia [11], 10.5% (64/609) in symptomatic and asymptomatic MSM in Sweden [41], and 10.3% (260/2510) in MSM in a community-based health centre in Lisbon [42], and it appears similar to rates reported in MSM attending a sexual clinic in India [7.2% (13/180)] [19] and MSM attending a clinic in South Africa [5.5% (5/91)] [43]. The prevalence in heterosexual men appears similar to gbMSM prevalence (14.6%) [44]. More data are required nationally and internationally.

Canadian trends showed variable reports of symptomatic infections (5–48%) and asymptomatic carriage (52% to 91.7%). Recently, Dumke et al. found asymptomatic *M. genitalium* carriage among 37.5% of gbMSM and 30.8% of heterosexual individuals. The asymptomatic carriage is highly variable by geographic area: low (0.58%) in France [45] and the United Kingdom (4.5%) [46]; middle in the United States (20.5%) [14]; and higher rates (93%) in Zurich [13]. The true prevalence and incidence of *M. genitalium* are unknown since current estimates are affected by variable access (e.g., availability or affordability) to testing and treatment in clinical settings as well as included populations in studies.

The clinical guidelines do not recommend *M. genitalium* screening in asymptomatic people. Ring et al. found that 30% of spontaneous bacterial clearance occurs in asymptomatic and symptomatic patients [13]. However, it is important to investigate the prevalence and risk factors associated with asymptomatic carriage to understand the consequences of asymptomatic infections and transmission as well as the impact of untreated asymptomatic infections.

Canadian data disaggregated by ethnicity is almost nil, with only one Canadian study examining ethnicity and reporting a higher prevalence among Indigenous people than among persons of non-Indigenous ethnicity [29]. Global *M. genitalium* epidemiology also reports disproportionately high prevalence among Black/African American people [14,15,40,47]. Canadian prevalence studies on other STIs also found that Indigenous and African American persons experience disproportionately higher rates of infection. [48,49]. The province of Manitoba recently reported the highest rates of HIV infection since the first case was reported in 1985 [50] and high rates of co-infection with other STIs. Infections were disproportionately among people who are consistently marginalized by the existing healthcare systems, are poorly housed, and inject drugs. Many self-identify as Indigenous, and many are young women [50]. These data highlight the need to incorporate ethnicity and other equity indicators to identify the most at-risk populations in persons with *M. genitalium* infection and to understand why.

*M. genitalium* coinfections are broadly reported with *C. trachomatis* and *N. gonorrhoeae*. The prevalence of coinfections is highly variable and also depends on the specimen types and diagnostic tests used [10,28,51]. Higher rates of coinfections with *C. trachomatis* (28.2%) and *N. gonorrhoeae* (23.8%) were recently reported, and the *C. trachomatis* coinfection was significant even after adjusting for specimen type [52]. Parmar et al. found that the prevalence of *M. genitalium* infection was significantly higher in females with *C. trachomatis* or *C. trachomatis*/*N. gonorrhoeae* coinfection and in males with *N. gonorrhoeae* [32]. Lê A et al. reported high rates of *M. genitalium* rectal coinfections in gbMSM in Montreal, 9.1% and 16.7% with *C. trachomatis* and *N. gonorrhoeae*, respectively [34]. The pathogenic implications of coinfections need further investigations to assess the potential effect on the reproductive health of females and males. For example, Scoullar et al. found greater reductions (−540.3 g, 95% CI: −859.3 to −221.2, *p* = 0.001) in the birthweight of newborns from females co-infected with *M. genitalium*, *C. trachomatis,* and *N. gonorrhoeae* compared to uninfected females [53]. In addition, in a systematic review, Frenzer et al. found *M. genitalium* as a probable risk factor of pre-term birth and spontaneous abortion [54]. Considering the impact of *M. genitalium* among young females, future studies are required to define the utility of *M. genitalium* testing in women of reproductive age.

The Canadian STI guidelines recommend *M. genitalium* testing from first-void urine; cervical, vaginal, urethral, or meatal swabs; and/or endometrial biopsies [23]. International guidelines recommend vaginal swabs for females and first-void urine in males [5,7,9]. Canadian studies comparing specimen types suggest cervicovaginal swabs perform somewhat better than urine to diagnose *M. genitalium* in women (15.3% vs. 12.6%, *p* = 0.035, respectively) [28], and more comfort using self-collected samples (urine and vaginal swabs) than those performed in a clinic [31]. Anal samples are only indicated in symptomatic patients, as the evidence shows no clear correlation between carriage and disease [8,55]. Pharyngeal testing is currently not recommended [5,7,9].

Canadian trends confirm the alarming reports of rising rates of antibiotic resistance in *M. genitalium*. In our study, the rates before 2017 ranged between 25 and 58%; however, after 2019, the macrolide resistance rates ranged between 53.6% and 82.1%, which is similar to other international trends [10,13,52]. These rates were even higher in some key populations [11,12,34,47,56]. The worldwide macrolide resistance has increased dramatically over the last few years. Machaleck et al. described an increase from 10% to 51% from 2010 to 2017, especially in the Western Pacific and the Americas regions [16]. Recently, the same author reported a slight, but not significant, decline in macrolide resistance prevalence in 2018–2021 (42% in 2015–2017 vs. 33% in 2018–2021 [18]). However, macrolide resistance prevalence is variable even between neighbouring countries [Denmark (9.0%), Norway (9.8%), and Sweden (7.2%)] [57].

Fluoroquinolone resistance is less common in Canada (2–29%), similar to other jurisdictions [16,21,47,58,59]. The rates before 2017 ranged from 0.0% to 20.0%, but after 2017 the fluoroquinolone resistance rates have reached up to 29%. Currently the rate of cure has decreased from 100% to 89% after 2010 [16], and some regions, such as the Asia-Pacific region, have a high level of fluoroquinolone resistance mutations in *M. genitalium* from men with symptomatic urethritis [60]. In addition, alarming data of fluoroquinolone-resistant strains in Berlin and Germany were reported by Glaunsinger and Dumke, showing an increase from 6.8% in 2017 to 38% in 2023 [12].

Macrolides (azithromycin) have been the preferred first-line antimicrobials for the treatment of *M. genitalium*, with fluoroquinolones, typically moxifloxacin, reserved for the treatment of macrolide-resistant strains. However, with the increasing resistance to first- and second-line antibiotics and the detection of dual resistance, recent guidelines and experts now recommend resistance-guided, sequential antimicrobial therapy whenever possible [7,9]. Data have shown that doxycycline decreases the *M. genitalium* organism load and, when combined with azithromycin, improves symptoms and results in higher rates of cure when used as part of a course of resistance-guided sequential therapy [61]. Treatment failures have been reported in patients infected with dually resistant strains; no other highly effective treatments are available. Pristinamycin results in about 75% cure and minocycline around 70% cure [62,63]. There is therefore an urgent need to explore additional therapeutic options for treatment and to revise current guidelines.

## 5. Conclusions

Canada does not conduct routine public health surveillance of *M. genitalium*. Current data were compiled from several cross-sectional studies in varying settings, populations, and geographic regions in Canada. *M. genitalium* prevalence and antimicrobial resistance patterns vary by geographic region and key population and show high circulating levels of resistance to macrolides and fluoroquinolones. The absence of population-based estimations of *M. genitalium* infections limits the understanding of its real impact, natural history and progress, drivers of antibiotic resistance, and disproportionate impact on key populations.

## Figures and Tables

**Figure 1 tropicalmed-10-00139-f001:**
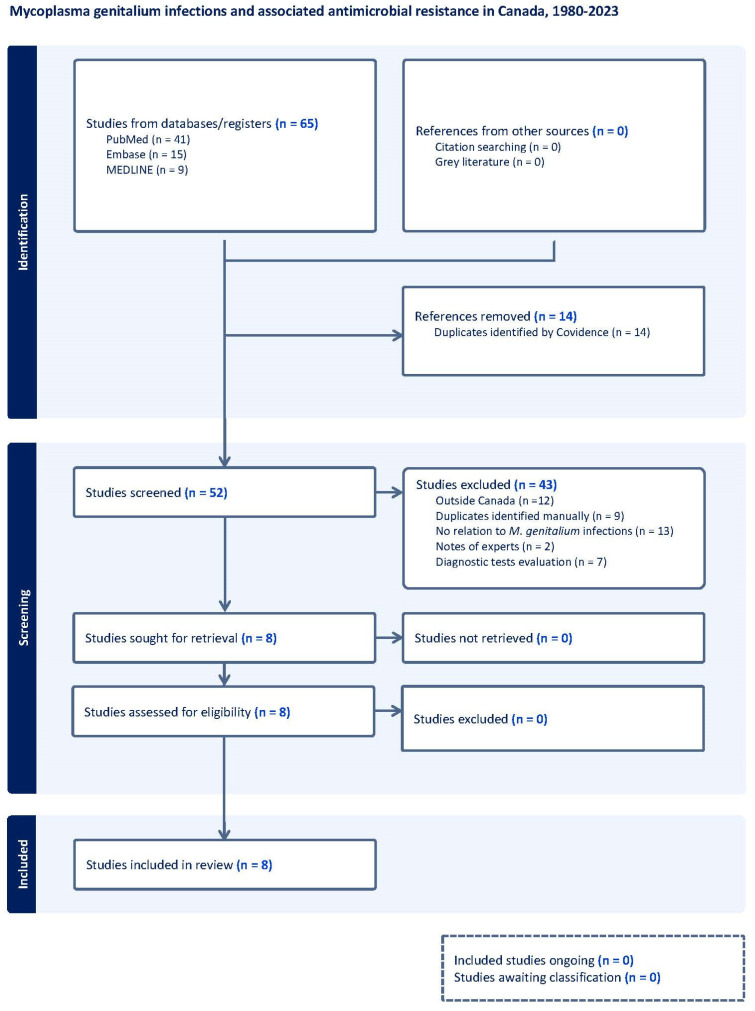
Flow diagram of studies included in the scoping review of the prevalence/frequency of *Mycoplasma genitalium* in Canada.

**Table 1 tropicalmed-10-00139-t001:** Population, province, prevalence/frequency of *Mycoplasma genitalium,* and diagnostic methods used in the studies conducted in Canada.

Authors	Timeframe of the Study	Population and Place of Recruitment	Province	Diagnostic Test Used to Detect *M. genitalium*	Prevalence/ Frequency % (*M. genitalium* Diagnosis/Total Sample)	Co-Infections with Other STIs Among People Diagnosed with *M. genitalium*
Gesink D. et al.	September 2013–December 2013	Males, females, and transgender clients seeking sexual health services	Ontario	Real-time PCR of MgPa gene	Global: 4.2% (50/1193) Women: 3.2% (10/309) Men: 4.5% (40/884)	Women: MG/CT: 0.3% (1/10) Men: MG/CT/NG: 0.1% (1/884) MG/NG: 0.5% (4/884)
Chernesky M. et al.	2015	Asymptomatic men in the clinic attendees	British Columbia	Aptima Mycoplasma genitalium assay RUO (Hologic, Marlborough, MA, USA) and Alt Amp targets	15.3% (54/354)	MG/CT: 22% (12/54) MG/NG: 1.9% (1/54) MG/TV: 1.9% (1/54) MG/HRHPV: 7.4% (4/54)
Gratrix J. et al.	January 2016–April 2016	Males and females seeking services in STI clinics	Alberta	Aptima Mycoplasma genitalium assay RUO (Hologic)	Global: 6.2% (142/2254) Females: 7.2% (75/1042) Males: 5.3% (64/1212) MSM: 6.6% (79/1212)	Female: MG/CT: 24% (18/75) MG/NG: 8% (6/75) MG/CT/NG: 4% (3/75) Male: MG/CT: 14.1% (9/64) MG/NG: 7.8% (5/64) MG/CT/NG: 15.3% (9/64)
Chernesky M. et al.	2017	Women seeking care for STIs	Ontario	Aptima Mycoplasma genitalium assay RUO (Hologic) and TMA MG laboratory-derived test (LDT)	5% (10/200)	MG/CT: 90% (9/10)
			British Columbia		11% (11/100)	MG/CT: 81.8% (9/11)
			Alberta		7.4% (15/202)	MG/CT: 60% (9/15)
			Saskatchewan		20% (20/100)	MG/CT: 80% (16/20)
			Manitoba		16% (16/100)	MG/CT: 50% (8/16)
			Quebec		3% (3/100)	MG/CT: 66.7% (2/3)
Lê A. et al.	November 2018 and November 2019	Gay, bisexual, and other men who have sex with men in a network of community-based research	Quebec	Allplex TM CT/NG/MG/TV assay (Seegene Inc., Seoul, Republic of Korea)	5.7% (41/716)	MG/CT: 4.3% (1/23, urethral swab) and 4.9% (2/41, rectal swab) MG/NG: 4.9% (2/41)
Chernesky M. et al.	2019	Women seeking care in STI clinics	Ontario	Aptima Mycoplasma genitalium assay RUO (Hologic)	30.3% (60/198)	MG/CT: 38.3% (23/60) MG/NG: 3.3% (2/60)
Parmar N. et al.	January 2019 and March/April 2019	Samples from females and males from the provincial laboratory	Saskatchewan	Aptima Mycoplasma genitalium assay RUO (Hologic)	Global: 9.6% (189/1977) Females: 11% (151/1368) Males: 6.2% (38/609)	Females: MG/CT: 12.6% (19/151) MG/NG: 1.3% (2/151) MG/CT/NG: 3.9% (6/151) Males: MG/CT: 13.2% (5/38) MG/NG: 10.5% (4/38) MG/CT/NG: 2.6% (1/38)
Chernesky M. et al.	2020	Asymptomatic and symptomatic women seeking care in a STIs clinic	Ontario	Seeplex STD6 ACE and ResistancePlus MG and Aptima MG	19.7% (59/300)	MG and/or CT/NG: 6.8% (4/59)

STIs: sexually transmitted infections. In this table, we report sex (female and male), gender (women, men, gender diverse), and sexual orientation (gay, bisexual, men who have sex with men, heterosexual, other) as the authors of the original study reported. MG: *Mycoplasma genitalium*. CT: *Chlamydia trachomatis*. NG: *Neisseria gonorrhoeae*. HRHPV: high-risk human papillomavirus. STIs: sexually transmitted infections.

**Table 2 tropicalmed-10-00139-t002:** *Mycoplasma genitalium* positivity rate reported in the studies, based on the specimen type used for diagnosis.

*M. genitalium* Prevalence According to the Specimen Type Used for Detection					
Authors	Urine % (n/N)	Cervical or Vaginal Swabs % (n/N)	Urethral Swabs % (n/N)	Rectal Swabs % (n/N)	Pharyngeal Swabs %(n/N)
Gesink D. et al.	4.2% (50/1193)	--	--	--	--
Chernesky M. et al.	12.6% (45/356)	--	15.3% (54/354)	--	--
Gratrix J. et al.	6.9% (25/362)	7.1% (26/362)	--	--	--
Chernesky M. et al.	11.4% (43/376)	7.6% (32/420)	--	--	--
Parmar N. et al.	9.6% (189/1977)	--	--	--	--
Lê A. et al.	--	--	2.20% (23/687) *	4.00% (41/688) *	0.2% (2/688) *
Chernesky M. et al.	26.7% (53/198)	28.7% (57/198)	--	--	--
Chernesky M. et al.	7.1% (21/297), 10.4% (31/297), and 17.2% (51/297) **	11% (33/300), 16% (48/300), and 19.7% (59/300) **	--	--	--

NT: Not tested. * Data from the original publication. Prevalence adjusted for respondent-driven sampling recruitment and censoring. ** Prevalence varied according to the *M. genitalium* technique used for the diagnosis: Seeplex STD6 ACE, ResistancePlus MG, and Aptima MG.

**Table 3 tropicalmed-10-00139-t003:** *Mycoplasma genitalium* resistance reported in Canada.

Authors	Timeframe of the Study	Population	Province	Macrolide Resistance Frequency	Fluoroquinolone Resistance Frequency
Gesink D. et al.	September 2013–December 2013	Male, female, and transgender clients seeking sexual health services	Ontario	58% (29/50)	20.0% (10/50)
Chernesky M. et al.	2015	Asymptomatic male clinic attendees	British Columbia	55.8% (19/34)	NT
Gratrix J. et al.	January 2016–April 2016	Males and females seeking services in STI clinics	Alberta	56.5% (52/92)	12.2% (5/41) *
Chernesky M. et al.	2017	Women seeking care for STIs	Ontario	50.0% (3/6)	16.7% (1/6) **
			British Columbia	25.0% (2/8)	0.0% (0/8) **
			Alberta	57.1% (8/14)	0.0% (0/14) **
			Saskatchewan	80% (8/10)	0.0% (0/8) **
			Manitoba	28.6% (4/14)	0.0% (0/14) **
			Quebec	33.3% (1/3)	0.0% (0/3) **
Lê A. et al.	November 2018 and November 2019	Gay, bisexual, and other men who have sex with men (gbMSM) in a network of community-based research	Quebec	82.1% (46/56)	29.1% (16/55) *
Chernesky M. et al.	2019	Women seeking care in an STI clinic	Ontario	74.4% (29/39)	NT
Parmar N. et al.	January 2019 and March/April 2019	Samples from females and males	Saskatchewan	63.6% (70/110)	10.5% (9/85) ***
Chernesky M. et al.	2020	Asymptomatic and symptomatic women seeking care in a STI clinic	Ontario	53.6% (22/41) for vs. and 48.0% (12/25) for FVU	NT
Institut National de Santé Publique du Québec	2018–2020	Samples from women and men from the provincial laboratory	Québec	80% (505/631)	22% (133/604)

NT: not tested. VS: vaginal swabs. FVU: first void urine. * No mutations detected in *gyrA* gene. ** Mutations in *gyrA* or *parC* genes. *** Mutations in *gyrA* gene were not analyzed.

## Data Availability

The epidemiological and laboratory data used for our analysis is publicly available in databases.

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
