# Peer review of "Mycoplasma genitalium Infections and Associated Antimicrobial Resistance in Canada, 1980–2023"

_tropicalmed, 2025, doi:10.3390/tropicalmed10050139_

Round 1

Reviewer 1 Report

Comments and Suggestions for Authors

See attachment

Author Response

Dear reviewer thank you so much for your suggestions and comments. The paper improved significantly thanks to the excellent comments from the three reviewers. We truly appreciate the time each one of you took to provide very specific suggestions.

Comments:

  1. This manuscript summarizes available data regarding the prevalence of Mycoplasma genitalium infection in Canada and the detection of macrolide and fluoroquinolone resistance markers. The authors searched a wide variety of sources and identified seven relevant studies. The results of those studies are compared in terms of prevalence of M. genitalium infection by specimen type, Canadian province, and antimicrobial resistance. The authors report important conclusions – that M. genitalium prevalence and antimicrobial resistance can be quite high in certain populations, and that more surveillance data is needed to guide testing and treatment in Canada. The manuscript could be improved by adding more details about the study populations, including statistical analysis in their comparisons, and making the discussion more concise. Specific comments: Line 41: specimen type (not types)…followed by cervical/vaginal swabs.

Answer: Thank you for the suggestion. We amended it accordingly.

  1. Line 41-43: Suggest to revise for clarity: “Coinfection with Chlamydia trachomatis was more common than coinfection wtih Neisseria gonorrhoeae...” Add statistics for that comparison and throughout – the goal is to draw conclusions from the aggregate data, which can only be done if significance is achieved (e.g., p<0.05).

Answer: Thank you for the suggestion. We appreciate the suggestion to perform a meta-analysis, however we did not define a priori due to the heterogeneity of the studies in terms of population, the inclusion and exclusion criteria, the differences in the specimen type and diagnostic test used.

  1. Line 46: Is this recommended by the Canadian Health organizations?

Answer: Yes, it is recommended by the Public Health Agency of Canada. According to the Canadian guidelines for STIs: “M. genitalium routine screening it is not recommended. Testing for M. genitalium is recommended only when chlamydia and gonorrhea have been ruled out as a cause of persistent or recurrent urethritis, cervicitis or pelvic inflammatory disease (PID) following empiric treatment for gonorrhea and chlamydia, when pre-treatment NAAT tests or follow-up TOC are negative for chlamydia and gonorrhea).”

https://www.canada.ca/en/public-health/services/infectious-diseases/sexual-health-sexually-transmitted-infections/canadian-guidelines/mycoplasma-genitalium/screening-diagnostic-testing.html

  1. Line 58-59: Suggest to revise: “…reported for women’s reproductive health. The impact of chronic infection in males is less clear.

Answer: Thank you for the suggestion. We amended it as you suggested.

  1. Line 60: “…due to the lack of regular testing…”

Answer: Thank you for the suggestion. We amended it accordingly.

  1. Line 71: Suggest to revise: “Initially recommend as the first line of treatment,

macrolids are now threatened…”

Answer: Thank you for the suggestion. We re write the sentence to have better clarity.

  1. Line 74: “differing prevalence in some geographic areas and some key populations”. Please expand – which populations are high vs low?

Answer: Thank you for the suggestion. We added the suggested information.

  1. Line 78: MG tests differ in their sensitivity. Were di\erent tests used in these studies? Which ones? Could that affect prevalence numbers?

Answer: Thank you for the questions. We included the test used in each study for M. genitalium detection in Table 1. And yes, the diagnostic test may affect the prevalence due to different sensitivities and specificities.

  1. Line 85: “Recovery”? Is “reviewing” a better word here?

Answer: Thank you for the suggestion. We amended it accordingly.

  1. Line 92: “sampling distribution”. Is this sample type? (ie, urine vs swab, etc)

Answer: Thank you for the suggestion. We amended it accordingly.

  1. Line 130: Suggest to revise “Over the 43 years….” with “We did not find any reports of surveillance data in the time frame of this study (1980-2025).

Answer: Thank you for the suggestion. We amended it accordingly.

  1. Table 1 and throughout: The patient population/setting should be noted for these studies. As the authors note, prevalence varies quite a bit depending on the population studied. Eg sexual health clinic, family medicine, seeking care for STI symptoms vs other indications, etc. Noting the province here is redundant with Table 3. Sample size is redundant with prevalence column. Is there a typo for Gratrix et al – 2254 vs 2294? NR: Not reported – I don’t see any NRs in this table…

Answer: Thank you for the suggestion! We fixed the mistakes as you suggested, deleted the sample size column and described in detail the population setting in Table 1. We also reported the prevalence by sex or gender according what the authors reported in each article.

  1. Line 144: “…the last study estimated…”

Answer: Thank you for the suggestion. We rewrite the sentence to have better clarity.

  1. Line 146: Omit “regardless of gender”?

Answer: Thank you for the suggestion. We rewrite the sentence to have better clarity.

  1. Line 151: Should this be: “Four studies reported prevalence with regard to symptoms.” Would this include symptoms and signs? Which symptoms? Urethritis, cervicitis?

Answer: Thank you for the suggestion. We rewrite the sentence to have better clarity. We added the symptoms description in those papers that reported them by women and men.

  1. Lines 163-166: Can anything else be added about these prevalences? Ie, men vs women, symptomatic vs asymptomic, patient population? It’s not very meaningful to just list them without other data. Perhaps a table?

Answer: Thank you for the suggestion. Using the information reported, we included additional information about the coinfections in table 1 as reported by each paper.

  1. Line 165: “four di\erent studies”

Answer: Thank you for the suggestion. We amended it accordingly.

  1. Line 169: “The most common specimen types…”

Answer: Thank you for the suggestion. We amended it accordingly.

  1. Table 2: If the studies are assigned a number that should be used in all of the tables.

Answer: Thank you for the suggestion. We amended it accordingly to have better consistency

  1. Unnecessary to include the references in each of the column headings. NT: Not tested.

Answer: Thank you for the suggestion. We amended it accordingly to have better consistency and fixed the typo mistake

  1. Line 182: Please list the parC and gyrA mutations detected as this would be of great interest to readers tracking resistance development worldwide. The mutations responsible for fluoroquinolone resistance/treatment failure are not well understood.

Answer: Thank you for the suggestion. We made and additional table included in the supplementary material (Table S2) that describe the detailed prevalence of 23SrRNA, parC and gyrA mutations listed in the papers, when this information was described.

  1. Table 3: This table is too busy. Suggest to make a column of study author (or study number as in Table 2) and remove references so that the Macrolide resistance and fluoroquinolone resistance columns show only numerical values. What is the di\erence between “Not detected” and 0%? This may be a good place to emphasize study population/setting as resistance is more common among eg. MSM. Typos in legend: NT: Not tested, ND = Not detected. ***, does this mean that gyrA mutations were not analyzed?

Answer: Thank you for the suggestion. Table 3 was changed as you suggested to made it simpler.

  1. Discussion Quite long and wide ranging. It should be made more concise and focused on the conclusions of this particular study.

Answer: We revised the discussion to make it more focused. Thank you.

  1. Lines 192-193: Confusing sentence… Suggest to revise:’ No routine surveillance information or data describing national population prevalence of genitalium were identified in our systematic search.”

Answer: Thank you for the suggestion. We amended it accordingly.

  1. Line 194: “…varies across the provinces…”. Or is it varying by study population/type? Can you support this statement with statistics? Are there enough studies to support the conclusion that there are geographic di\erences?

Answer: Thank you for the suggestion. We wrote again the sentence to make it cleared. In addition, Table 3 shows the prevalence distribution of macrolide and fluoroquinolone resistance strains across the Canadian provinces included in our analysis. Because there are very low number of papers with very heterogenous criteria, a statistical analysis would not be appropriate.

  1. Line 207: “differing geographic prevalence”. Again, is it geography or study population? Or something else?

Answer: Thank you for the suggestion. We wrote again the sentence to make it clearer. The differences were presented in both senses, by provinces and by the population included in the studies.

  1. Lines 225-226: Awkward sentence...suggest to revise: “Comparing all of the Canadian studies we found that genitalium positive females were younger (median or mean 25-27 years old) than M. genitalium positive males (median or mean 30-32 years old).” Needs statistics.

Answer: Thank you for the suggestion. We amended it accordingly.

  1. Line 247: The influence of specimen type on M. genitalium load quantification should be mentioned as that is a major conclusion of the Munson study. The relevance of this information to the current study isn’t clear if the specimen type isn’t described. For example, can the authors say that one study had lower MG prevalence because they used a less sensitive specimen (or test)?

Answer: Thank you for the suggestion. We amended it accordingly.

  1. Line 267: Which geographic areas correspond to those numbers?

Answer: Thank you for the suggestion. We amended it accordingly.

  1. Line 271: “The role of symptomatic infection and its sequelae remains unknown.”

Answer: Thank you for the suggestion. We amended it accordingly.

  1. Line 276: Unsure what this sentence means. Are the authors saying that only one Canadian study examined ethnicity?

Answer: Thank you for the suggestion. We amended it accordingly.

  1. Lines 283-284: Relevance of pap smear access to the current study?

Answer: Thank you for the suggestion. We changed the focus of this sentence and restructured the information according to the study Scopus.

  1. Line 287: “…or hepatitis C coinfections.”

Answer: Thank you for the suggestion. We amended it accordingly.

  1. Lines 302-303: Are they suggesting that all women of reproductive age should be screened for MG? Is there data to support this approach given the difficulty in treating MG?

Answer: Thank you for the suggestion. We amended it accordingly.

  1. Lines 306-307: Omit parentheses.

Answer: Thank you for the suggestion. We amended it accordingly.

  1. Lines 307-309: Reword: “Canadian studies comparing specimen type suggest that swabs (what kind?) perform somewhat better than urine to diagnose M. genitalium in women….”

Answer: Thank you for the suggestion. We amended it accordingly.

  1. Line 315: Revise to: “had macrolide resistance mutations”

Answer: Thank you for the suggestion. We amended it accordingly.

  1. Line 321: Give examples of differences in prevalence

Answer: Thank you for the suggestion. We included this information as requested.

  1. Line 326-328: “…who showed that the prevalence of fluoroquinolone resistance increased from 6.8% in 2017 to 38% in 2023…”

Answer: Thank you for the suggestion. We amended it accordingly.

  1. Lines 336-338: Revise: “Treatment failures have been reported in patients infected with dually resistant strains; no other highly effective treatments are available.

Answer: Thank you for the suggestion. We amended it accordingly.

Reviewer 2 Report

Comments and Suggestions for Authors

This is an interesting study focusing on the prevalence and antimicrobial resistance patterns of Mycoplasma genitalium across Canada. I have several comments which could improve the manuscript: 

Line 60 = “due to the regular of” – please amend

Lines 61 – 63 – you can also note that the prevalence seems to be higher than gonorrhoea and is similar to that of chlamydia, based on the available literature.

Line 70 – italics for M. genitalium

Line 71 – 72 – the global average for macrolide resistance according to Machalek is ~50% across most urban centres globally.

Line 73-75 – fluoroquinolone resistance estimates are ~7% globally, based on Machalek et al., it would be good to provide some examples as to the prevalence of FQR in different geographic areas.

Lines 78-80 – can you please clarify is this testing simply for diagnosis of MG or related to antimicrobial resistance testing (or both)?

Table 1 – “Chlamydia Trachomatis” – please remove capital letter for trachomatis

Technical query related to table 1 – would it be possible to detail the method used to detect MG. I realise that these will all be PCR, but some assays are more sensitive than others and I would be interested to see the various methods used here. Also, the paper looking at asymptomatic men – was this all men (i.e. both hetero and GBMSM)?

Line 144 – “The last study estimate the prevalence” – please amend the grammar here

Can you please report the studies by the first author surname, rather than the “Alberta study” – this is because there is data from Alberta in another study, so doing this will avoid confusion.

Regarding the studies with co-detection of other STIs, can you please further comment on what patient populations these related to. Obviously there is a wide spread of % co-infections among the studies, but was this potentially related to patient populations (e.g. HIV, MSM etc)

Line 169 – “the most specimen types were” – please amend the grammar here

Table 3 – I think you need to break this down further, especially for fluoroquinolone resistance. Specifically, it would be good to understand and/or report the variety of macrolide resistance mutations (e.g. how many were A2058G, how many were A2059G etc). The same should be done for fluoroquinolones, noting studies where specific testing for gyrA was not performed, and also based on the fact that available current literature (Sweeney et al 2022 10.1016/S1473-3099(21)00629-0; Murray et al. 2022 10.1128/aac.00278-22, Murray et al. 2023 10.1093/cid/ciad057, Ertl et al 2024 10.1093/jac/dkad373 and Chua et al 2025 10.1016/j.lanmic.2024.101047) demonstrates that the only ParC mutation that is linked to clinical treatment failure is ParC-S83I (G248T), while other parC mutations are of limited clinical utility. Similarly, in samples with ParC-S83I and a concurrent GyrA mutation (any mutation, based on available but limited literature), the risk of treatment failure with fluoroquinolones (both moxi and sita) is further increased. Please revise this table accordingly to provide a breakdown of the types of mutations observed in these studies, or at a minimum this should be provided as a supplemental table.

Line 220 – I think this should be phrased as African American (not black).

Can you please provide a supplemental table detailing the differences in MG by sample type and also include the type of PCR assay they used to determine MG diagnosis, as differences in the molecular diagnostic assay used may differ in their capacity to detect MG in different sample types. 

Line 255-260 – but what are the numbers of patients like included in these studies vs your Canadian data? Yes, rates will of course differ, but is the Canadian dataset based on a much smaller number of samples vs others? You say on line 261 that data frm Canadian studies is scares, but some context here would be great.

Line 271 – do you mean sequelae?

Line 275 – please amend the grammar here (“the impact of no treated asymptomatic infections”)

Line 280 – should be “STIs” (not STI), also please refer to this as African American (not black)

Lines 299 – 303, you mention one specific study related to MG infection and birthweight of babies; there is an entire systematic review on the role of MG in pregnancy complications and this article should be mentioned and referenced here (https://pubmed.ncbi.nlm.nih.gov/35351816/)

Discussion lines 314-321 – there is now a revised systematic review and meta-analysis of MG AMR (Chua et al: https://pubmed.ncbi.nlm.nih.gov/40147462/) it would be worthwhile referencing this article here.

Lines 334-336 – while it is true that doxycycline seems to lower the MG load, I think it is also pertinent to state that doxycycline alone cannot cure the majority of MG infections (see Jorgen Jensen articles, citing ~two thirds of MG cases will not be cured with doxy alone).

Comments on the Quality of English Language

Some minor grammar and typographical errors. Please carefully check the manuscript 

Author Response

Dear reviewer, thank you so much for your excellent and specific suggestions. As we made extensive revisions thanks to the three reviewers, some of the lines you initially cited in your comments may have changed in the revised version.

Comments:

  1. This is an interesting study focusing on the prevalence and antimicrobial resistance patterns of Mycoplasma genitalium across Canada. I have several comments which could improve the manuscript: Line 60 = “due to the regular of” – please amend

Answer: Thank you for the suggestion. We amended it accordingly.

  1. Lines 61 – 63 – you can also note that the prevalence seems to be higher than gonorrhoea and is similar to that of chlamydia, based on the available literature.

Answer: Thank you. We rewrote the sentences.

  1. Line 70 – italics for  genitalium

Answer: Thank you for the suggestion. We amended it accordingly.

  1. Line 71 – 72 – the global average for macrolide resistance according to Machalek is ~50% across most urban centres globally.

Answer: Thank you for the suggestion. We amended it accordingly.

  1. Line 73-75 – fluoroquinolone resistance estimates are ~7% globally, based on Machalek et al., it would be good to provide some examples as to the prevalence of FQR in different geographic areas.

Answer: Thank you for the suggestion. We included additional examples as you suggested.

  1. Lines 78-80 – can you please clarify is this testing simply for diagnosis of MG or related to antimicrobial resistance testing (or both)?

Answer: Thank you for the recommendation. We clarified that is for both.

  1. Table 1 – “Chlamydia Trachomatis” – please remove capital letter for trachomatis

Answer: Thank you for the suggestion. We amended it accordingly.

  1. Technical query related to table 1 – would it be possible to detail the method used to detect MG. I realise that these will all be PCR, but some assays are more sensitive than others and I would be interested to see the various methods used here. Also, the paper looking at asymptomatic men – was this all men (i.e. both hetero and GBMSM)?

Answer: Thank you for the suggestion. We added the diagnostic test used to detect M. genitalium reported by each study in the Table 1. We also added the prevalence of M. genitalium by sex or gender, based on what the authors reported in each study.

Regarding your question about the asymptomatic men for that particular study, unfortunately, the article did not specify the participants’ sexual orientation in the study population descriptions and the results.

  1. Line 144 – “The last study estimate the prevalence” – please amend the grammar here

Answer: Thank you for the suggestion. We amended it accordingly.

  1. Can you please report the studies by the first author surname, rather than the “Alberta study” – this is because there is data from Alberta in another study, so doing this will avoid confusion.

Answer: Thank you for the suggestion. We edited the three tables to report the first author of each article.

  1. Regarding the studies with co-detection of other STIs, can you please further comment on what patient populations these related to. Obviously there is a wide spread of % co-infections among the studies, but was this potentially related to patient populations (e.g. HIV, MSM etc)

Answer: Thank you so mush for your suggestion. As we mentioned earlier, we added what the patient population to table 1, and with this comment we added the coinfections reported by each paper. We also added additional details about coinfections in particular groups, and commented about it in the discussion, lines 320-332.

  1. Line 169 – “the most specimen types were” – please amend the grammar here

Answer: Thank you for the suggestion. We amended it accordingly.

  1. Table 3 – I think you need to break this down further, especially for fluoroquinolone resistance. Specifically, it would be good to understand and/or report the variety of macrolide resistance mutations (e.g. how many were A2058G, how many were A2059G etc). The same should be done for fluoroquinolones, noting studies where specific testing for gyrA was not performed, and also based on the fact that available current literature (Sweeney et al 2022 10.1016/S1473-3099(21)00629-0; Murray et al. 2022 10.1128/aac.00278-22, Murray et al. 2023 10.1093/cid/ciad057, Ertl et al 2024 10.1093/jac/dkad373 and Chua et al 2025 10.1016/j.lanmic.2024.101047) demonstrates that the only ParC mutation that is linked to clinical treatment failure is ParC-S83I (G248T), while other parC mutations are of limited clinical utility. Similarly, in samples with ParC-S83I and a concurrent GyrA mutation (any mutation, based on available but limited literature), the risk of treatment failure with fluoroquinolones (both moxi and sita) is further increased. Please revise this table accordingly to provide a breakdown of the types of mutations observed in these studies, or at a minimum this should be provided as a supplemental table.

Answer: Thank you for the suggestion. We included an additional table in the supplementary material (table S2) that describes in detail the variety of macrolide and fluoroquinolone resistance mutations.

  1. Line 220 – I think this should be phrased as African American (not black).

Answer: Thank you for the suggestion. We amended it accordingly.

  1. Can you please provide a supplemental table detailing the differences in MG by sample type and also include the type of PCR assay they used to determine MG diagnosis, as differences in the molecular diagnostic assay used may differ in their capacity to detect MG in different sample types. 

Answer: Thank you so much for your suggestion. To clarify, we described the M. genitalium positivity by specimen type in Table 2. The positivity is reported by sample type according to the initial reports described by each author. The PCR assays used for the diagnosis were included in Table 1 accordingly.

  1. Line 255-260 – but what are the numbers of patients like included in these studies vs your Canadian data? Yes, rates will of course differ, but is the Canadian dataset based on a much smaller number of samples vs others? You say on line 261 that data frm Canadian studies is scares, but some context here would be great.

Answer: Thank you for the suggestion. We included the sample sizes and settings characteristics of recent studies developed in MSM outside Canada, currently lines 275-284.

  1. Line 271 – do you mean sequelae?

Answer: Thank you for the suggestion. We amended it accordingly.

  1. Line 275 – please amend the grammar here (“the impact of no treated asymptomatic infections”)

Answer: Thank you for the suggestion. We amended it accordingly.

  1. Line 280 – should be “STIs” (not STI), also please refer to this as African American (not black)

Answer: Thank you for the suggestion. We amended it accordingly.

  1. Lines 299 – 303, you mention one specific study related to MG infection and birthweight of babies; there is an entire systematic review on the role of MG in pregnancy complications and this article should be mentioned and referenced here (https://pubmed.ncbi.nlm.nih.gov/35351816/)

Answer: Thank you for the suggestion. We amended it accordingly and the reference was added.

  1. Discussion lines 314-321 – there is now a revised systematic review and meta-analysis of MG AMR (Chua et al: https://pubmed.ncbi.nlm.nih.gov/40147462/) it would be worthwhile referencing this article here.

Answer: Thank you for the suggestion. We amended it accordingly and the reference was added.

  1. Lines 334-336 – while it is true that doxycycline seems to lower the MG load, I think it is also pertinent to state that doxycycline alone cannot cure the majority of MG infections (see Jorgen Jensen articles, citing ~two thirds of MG cases will not be cured with doxy alone).

Answer: Thank you for the suggestion. We rewrite the sentence to have better clarity.

  1. Comments on the Quality of English Language. Some minor grammar and typographical errors. Please carefully check the manuscript 

Answer: Thank you for the suggestion. The entire article was proofread.

Reviewer 3 Report

Comments and Suggestions for Authors

Estimated Authors,

I've read with great interest your report on the occurrence of M genitalium infections in Canada, focusing on the associated AMR data (1980-2022). Interestingly enough, your study was unable to identify any institutional report on the assessed topic, suggesting the need (but I would say "the urgent need") for a better surveillance system. On the other hand, several reports published on peer-reviewed journals were published and accurately summarized.

Unfortunately, the lack of institutional data and the subsequent shifting of the paper from a comprehensive review to something alike a systematic review should recommend the Authors to also improve and re-modulate their reporting strategy.

More precisely: 

even without implementing a full PRISMA statement strategy, Authors should provide some degree of rating of the included studies in terms of their potential bias; moreover, Authors should provide some more detailed information about the search strategy (i.e. how many studies were initially retrieved, then removed from the analyses by providing the conventional flow-chart from systematic reviews). 

Please fix Table 2 by replacing the number of the study with the full reference in a consistent way with Table 1.

Please also consider whether a meta-analytical approach could further improve the overall quality and content of the present study.

Author Response

Dear reviewer thank you very much for your excellent comments and thanks to those suggestions the revised manuscript now incorporates a detailed description of what we did.

Comments:

  1. Estimated Authors, I've read with great interest your report on the occurrence of M genitalium infections in Canada, focusing on the associated AMR data (1980-2022). Interestingly enough, your study was unable to identify any institutional report on the assessed topic, suggesting the need (but I would say "the urgent need") for a better surveillance system. On the other hand, several reports published on peer-reviewed journals were published and accurately summarized.

Answer: Thank you so much for your kind comment.

  1. Unfortunately, the lack of institutional data and the subsequent shifting of the paper from a comprehensive review to something alike a systematic review should recommend the Authors to also improve and re-modulate their reporting strategy. More precisely: even without implementing a full PRISMA statement strategy, Authors should provide some degree of rating of the included studies in terms of their potential bias; moreover, Authors should provide some more detailed information about the search strategy (i.e. how many studies were initially retrieved, then removed from the analyses by providing the conventional flow-chart from systematic reviews). 

Answer: Thank you so much for your suggestion. We expanded on all methods we did but we did not report it previously. The revised manuscript included in our methods and results a complete description of the search strategy, databases and information about the number of papers retrieved, duplicates, the screening by title and abstract, and full-text reading. We included the flowchart of all the process in Figure 1. In addition, we included in the supplementary material the quality evaluation done by two blind and independent reviewers for the papers analyzed in the scoping review.

  1. Please fix Table 2 by replacing the number of the study with the full reference in a consistent way with Table 1.

Answer: Thank you for the suggestion. We fixed it accordingly in the three tables.

  1. Please also consider whether a meta-analytical approach could further improve the overall quality and content of the present study.

Answer: Thank you for the suggestion. We followed a systematic process and evaluation during the entire process, but we did not define a priory a meta-analysis approach. In addition, considering the high heterogeneity in study populations and variable inclusion and exclusion criteria, diagnostic tests and specimen type used in each study, and that all studies are cross-sectional, we considered that it is not appropriate to perform a meta-analysis.

Round 2

Reviewer 1 Report

Comments and Suggestions for Authors

Two comments:

(1) The flow diagram did not appear properly in the revised manuscript so it could not be evaluated.

(2) The discussion is still very long, it doesn't seem to have been shortened at all from the first submission.

Author Response

Dear Reviewer thank you for the comments. Please find below our answers to your comments:

(1) The flow diagram did not appear properly in the revised manuscript so it could not be evaluated.

Answer: We copied and pasted the Figure 1, which is the flowchart we downloaded from Covidence. In the previous resubmission, I attached a PDF with track changes and a clean version in the same file. When I checked the files you reviewed, I noticed the clean version was removed. I attached in this resubmission the clean version to avoid the track changes modify the flowchart. 

(2) The discussion is still very long, it doesn't seem to have been shortened at all from the first submission.

Answer: Dear reviewer, we cut at least 30% of the discussion in the previous resubmission and reorganized it completely to make it more cohesive. We also deleted additional paragraphs in the current resubmission. We had to add some suggestions from Reviewers 2 and 3 that we felt were amazing to show the contrast. We kept those suggestions from the reviewers 2 and 3. I attach the clean version in this submission in case reading a clean version facilitates the review of the latest revised version of the manuscript.

Thanks for your comments.